# On the Role and Production of Polyhydroxybutyrate (PHB) in the Cyanobacterium *Synechocystis* sp. PCC 6803

**DOI:** 10.3390/life10040047

**Published:** 2020-04-22

**Authors:** Moritz Koch, Kenneth W. Berendzen, Karl Forchhammer

**Affiliations:** 1Interfaculty Institute of Microbiology and Infection Medicine Tübingen, Eberhard-Karls-Universität Tübingen, 72076 Tübingen, Germany; moritz.koch@uni-tuebingen.de; 2Center for Plant Molecular Biology, Eberhard-Karls-Universität Tübingen, 72076 Tübingen, Germany; kenneth.berendzen@zmbp.uni-tuebingen.de

**Keywords:** cyanobacteria, bioplastic, PHB, sustainable, resuscitation, chlorosis, bacterial survival, *Synechocystis*, biopolymers

## Abstract

The cyanobacterium *Synechocystis* sp. PCC 6803 is known for producing polyhydroxybutyrate (PHB) under unbalanced nutrient conditions. Although many cyanobacteria produce PHB, its physiological relevance remains unknown, since previous studies concluded that PHB is redundant. In this work, we try to better understand the physiological conditions that are important for PHB synthesis. The accumulation of intracellular PHB was higher when the cyanobacterial cells were grown under an alternating day–night rhythm as compared to continuous light. In contrast to previous reports, a reduction of PHB was observed when the cells were grown under conditions of limited gas exchange. Since previous data showed that PHB is not required for the resuscitation from nitrogen starvation, a series of different abiotic stresses were applied to test if PHB is beneficial for its fitness. However, under none of the tested conditions did cells containing PHB show a fitness advantage compared to a PHB-free-mutant (Δ*phaEC*). Additionally, the distribution of PHB in single cells of a population *Synechocystis* cells was analyzed via fluorescence-activated cell sorting (FACS). The results showed a considerable degree of phenotypic heterogeneity at the single cell level concerning the content of PHB, which was consistent over several generations. These results improve our understanding about how and why *Synechocystis* synthesizes PHB and gives suggestions how to further increase its production for a biotechnological process.

## 1. Introduction

Cyanobacteria have colonized our planet for more than two billion years and are widespread within the light-exposed biosphere [1]. Their ability to conduct oxygenic photosynthesis enables them to survive under extreme environmental conditions, even in the absence of organic carbon sources. In adaption to these diverse environments, many cyanobacteria have evolved the ability to produce a variety of biopolymers [2]. Most of the mentioned polymers serve to store macro-nutrients, like carbon (in the form of glycogen), phosphate (polyphosphate), or nitrogen (cyanophycin). Although it has been known since 1966 that cyanobacteria also possess polyhydroxybutyrate (PHB), its function remains puzzling [3,4,5].

Around 30 years later, Hein et al. found that the required biosynthetic genes for the PHB synthesis are also present in the model organism *Synechocystis* sp. PCC 6803 (hereafter “*Synechocystis*” or “WT” for wild-type) [6]. Soon after, it has been shown that this strain is indeed capable of producing PHB under nutrient limited conditions [7]. It has been hypothesized that the polymer serves as an additional carbon and energy storage, similarly to glycogen, which could help to survive environmental stress conditions. However, until today, the true physiological function remains unknown [5].

In other organisms, PHB can fulfill manifold functions. The polymer often accumulates under nutrient limitation or unbalanced conditions (e.g., an excess of carbon) [8]. In certain organisms like *Ralstonia eutropha*, it is also accumulated during normal growth phase [9]. In the strain *Azospirillum brasilense*, for example, heat, UV irradiation, desiccation, osmotic shock, and osmotic pressure affect the growth of a PHB deficient strain [10]. In the strain *Herbaspirillum seropedicae*, PHB is able to reduce redoxstress. Hence, PHB could serve as an electron sink to eliminate a surplus of reducing equivalents [11]. A similar behavior has been shown in the anoxygenic phototrophic bacterium *Chromatium vinosum*: it converts glycogen to PHB (under anerobic, dark metabolism) [12]. Thereby, the strain does not lose carbon, compared to other bacteria, which secrete their fermentation products. In agreement, it has been shown that excess NADPH sustains PHB accumulation [13].

A better understanding about the production of PHB in cyanobacteria would be beneficial for the biotechnological production of the biopolymer. PHB from cyanobacteria is suggested as a sustainable source for biodegradable plastics [14]. However, the current production rates are rather low, making it difficult to commercially compete with the PHB production in heterotrophic organisms [15]. So far, most attempts to increase the PHB yield focused on either medium optimization or metabolic engineering approaches. At the same time, fundamental questions about the PHB metabolism and PHB-forming conditions have been neglected. For instance, it has just recently been discovered that PHB is mostly formed from intracellular glycogen [16,17]. This work therefore aims to better understand the conditions, under which PHB is produced in *Synechocystis*, as well as gaining further insights in the physiological function of PHB within the cyanobacterial metabolism.

This knowledge will be helpful for both basic and applied research, since *Synechocystis* is considered a promising host for the industrial production of bioplastic from PHB [18].

## 2. Materials and Methods

### 2.1. Cyanobacterial Cultivation Conditions

*Synechocystis* sp. PCC 6803 cells were grown in standard BG_11_ medium as described before [19]. Additionally, 5 mM NaHCO_3_ were added. All used strains are listed in Table A1. To ensure the preservation of the genetic modifications, appropriate antibiotics were added to the different mutant strains. All cells were pre-adapted to their growth conditions by growing a pre-culture for 3 days at the same condition. For normal growth, cells were grown under constant illumination of ~50 µE m^−2^ s^−1^ and at 28 °C. Aeration was provided by continuous shaking at 120 rpm. Either 50 or 200 mL bacterial culture were grown in baffle free Erlenmayer flasks. Whenever nitrogen starvation was required, cells were shifted to nitrogen depleted medium as previously described [20]. In short, 200 mL exponentially growing cells at an OD_750_ of ~0.8 were centrifuged at 4000 g for 10 min. The pellet was resuspended in 100 mL BG_0_ (BG11 without any sodium-nitrate) and centrifuged again. The pellet was then resuspended in BG_0_ once more and the OD_750_ was adjusted to 0.4. For resuscitation experiments, a chlorotic culture was spun down and resuspended in BG_11_ medium.

### 2.2. Physical Stress Conditions

To test whether the formation of PHB is advantageous under conditions of physical stress, chlorotic WT and Δ*phaEC* cells, which were starved from nitrogen for ~2 weeks, were used. The cells were treated with the conditions described in category 1 of Table 1 and subsequently recovered on BG_11_ plates. Afterwards, a serial dilution of cell suspension, from OD_750_ ~1 (= 100) until 10-4, was prepared. From each dilution, 5 µL were dropped on an BG11 agar plates containing 1.5 % agar and incubated for 1–2 weeks under continuous light, until visible colonies formed. Alternatively, chlorotic cells were recovered on BG_11_ plates with additional ingredients (category 2 of Table 1) which can cause stress, e.g., high salt concentrations. The number of formed colonies was compared between the WT and ΔphaEC cells.

### 2.3. Oxygen Measurements

To measure oxygen levels in a liquid culture, an oxygen detecting sensor was placed at the bottom of a standing culture. The readout was performed using an OXY-1 SMA device (PreSens, Regensburg, Germany). At the beginning of the measurements, *Synechocystis* cells were shifted to nitrogen free BG0 medium and the oxygen levels were monitored for three constitutive days. The equilibrium of dissolved oxygen within the cultures was measured at 360 µM/L.

### 2.4. Microscopy and Staining Procedures

To visualize cell morphology and PHB granules within the cells, 100 µL of cyanobacterial culture were centrifuged. The resulting pellet was resuspended in a mixture of 10 µL Nile red and 20 µL water. From the resuspended mixture, 10 µL were used and dropped on an agarose-coated microscopy slide. A Leica DM5500B microscope (Leica, Wetzlar, Germany) with a 100 × /1.3 oil objective was used for fluorescence microscopy. For the detection of Nile red stained PHB granules, a suppression filter BP 610/75 was used, together with an excitation filter BP 535/50. The pictures were taken by a Leica DFC360FX.

### 2.5. Electron Microscopy

For detailed pictures of the intracellular PHB granules, electron microscopy was used. For this, glutaraldehyde and potassium permanganate were used to fix and postfix *Synechocystis* cells. Citrate and uranyl acetate were used to stain ultrathin sections, as described before [21]. Finally, a Philips Tecnai 10 electron microscope (Philips, Amsterdam, Netherlands) was used to examine the samples at 80 kHz.

### 2.6. Spectral Analysis

To measure the whole-cell absorption spectrum, a Specord 50 with the software WinAspect (Analytik Jena, Jena, Germany) was used. The absorption was measured between 350 and 750 nm. The spectra were normalized to the OD_750_. To determine the recovery of the photopigments, the change in absorption between the induction of the recovery (t0) and after three days (t72) was determined (normalized to OD_750_). For this, the specific wavelengths for phycobilisomes and chlorophyll (absorption at 630 nm and 680, respectively) were compared.

### 2.7. PAM

To detect the photosynthetic activity, pulse–amplitude–modulation fluorometry (PAM) was used. This measures the relative quantum yield of the photosystem II, Y(II). A Heinz Walz GmbH (Effeltrich, Germany) WATER-PAM Chlorophyll Fluorometer with WinControl Software was used. For the measurements, a cell suspension with an OD_750_ between 0.4 and 1 was used and diluted 20-fold. After 5 min incubation in the dark, the maximum PSII quantum yield (Fν/Fm) was determined applying the saturation pulse method [22]. For each time point, three measurements with a time constant of 30 s were taken.

### 2.8. PHB Quantification

To determine the intracellular PHB content, ~15 mL of cells were centrifuged at 4000 g for 10 min. The pellet was dried for at least 2 h at 60 °C in a speed vac until all pellets were dry. Next, 1 mL of concentrated sulfuric acid (18 M H_2_SO_4_) were added and the mixture was boiled for 1 h at 100 °C. This process releases PHB from the cells and converts it to crotonic acid. From this, 100 µL were taken and diluted with 900 µL of 14 mM H_2_SO_4_. The sample was centrifuged for 5 min at 20,000 g. From the supernatant, 500 µL were transferred into a new tube and combined with 500 µL H_2_SO_4_. The samples were centrifuged once more for 5 min at max speed and 400 µL of the resulting supernatant was used for further HPLC analysis. For this, a Nucleosil 100 C 18 column was used (125 by 3 mm). For a liquid phase, 20 mM phosphate buffer with pH 2.5 was used. Crotonic acid was detected at 250 nm. As a standard, commercially available crotonic acid was used, with a conversion rate to PHB of 0.893.

### 2.9. FACS and Flow Cytometry

*Synechocystis* cells that were starved from nitrogen for ~2 weeks before FACS experiments. Cells were sorted with a MoFlo XDP (Beckman Coulter, Munich, Germany) into 500 µL PBS buffer using a 70 uM CytoNozzle at 30 p.s.i. and PBS [pH 7.0] as sheath. Before FACS or analysis, 1 µL of BODIPY (5 mg/mL) was added to 500 µL of cells and incubated for 10 min. Cells were identified based on their scatter (SSC-LA vs. FSC-LA), chlorophyll-fluorescence (670/30) captured from a 488 nm (70 mW) laser. Cells were divided into low and high producers based on their emission profile and at 534/30 (BODIPY) when compared to unstained and PHB deficient cells. For analysis, the software Summit FACS and FCS Express was used. BODIPY staining was also scored using a Cytoflex analyzer (Beckmann Coulter, Munich, Germany) equipped with a single 488 nm laser. Principle BODIPY emission was captured with a 525/40 bandpass and plotted against scatter to remove clumplets and distinguish BODIPY. PHB content was inferred when compared to unstained and PHB deficient cells. For analysis and illustration of the date, the software programs CytoExpert (Beckman Coulter, Munich, Germany) and FlowJo (FlowJo LLC, Oregon, USA) were used.

## 3. Results

### 3.1. The Influence of Environmental Conditions and Central Pathways on PHB Production

In PHB producing cyanobacteria, PHB synthesis is efficiently induced by depleting the nitrogen source from the medium. This is also the case for the model cyanobacterium *Synechocystis* sp. PCC 6803 used in this study. To further understand the conditions, under which PHB is produced, cells, which were transferred to nitrogen-depleted medium, were grown under different conditions of aeration and illumination (Figure 1).

It was shown in a previous study that the limitation of gas exchange can boost the PHB production [23]. To verify this observation, cells were grown without any shaking to create a situation of reduced aeration. Under these standing conditions, the cells produced oxygen and were oxygen-oversaturated during the day, whereas they consumed it during the night, resulting in transient periods of limited oxygen availability (Figure A2). Furthermore, cyanobacteria are naturally adapted to day-night rhythms, while they are commonly grown under continuous light in the laboratory. To test whether this affects the PHB production, PHB was quantified from cells grown under both light regimes. It turned out that, compared to standard laboratory conditions of continuous light and shaking, a limitation in gas exchange always led to a reduced PHB content. In contrast, the growth under day-night rhythm resulted, both in standing and shaken cultures, and in an increased content of PHB per cell-dry-weight (CDW). Therefore, the maximal PHB production was achieved in shaken cultures in a 12 h light/dark regime.

To test whether these effects are linked to a specific carbon pathway, knockout mutants of two central pathways were used: The mutant Δ*pfk* is unable to metabolize carbon via the the EMP (Embden–Meyerhof–Parnas) pathway and the mutant Δ*zwf* cannot use the OPP (oxidative pentose phosphate) and ED (Entner–Doudoroff) pathway. Both strains and a WT control were grown under the same conditions as in the previous experiment and their PHB content was then compared (Figure 2).

The Δ*zwf* mutant lacks glucose-6P dehydrogenase, which feeds sugar catabolites into the two central pathways ED and OPP, which are crucial for vegetative growth. These pathways are neglectable for the production of PHB under nitrogen starvation, since the Δ*zwf* mutant showed the same production pattern as the WT under all tested conditions (see Figure 2). In contrast, the Δ*pfk* mutant, which is unable to use the EMP pathway, showed a strongly impaired PHB production under all conditions. Interestingly, growth with a 12 h light/dark regime caused increased PHB production in the WT and the Δ*zwf* mutant, whereas it resulted in a severe reduction of PHB in the Δ*pfk* strain.

### 3.2. Recovery from Nitrogen Starvation

To test whether the formation of PHB plays a role for the recovery from nitrogen chlorosis under specific conditions, WT and a PHB-free mutant (Δp*haEC*) were nitrogen-starved for 18 days and were transferred to standard BG_11_ medium to induce resuscitation. Thereafter, parameters, which indicate the process of the resuscitation, such as the yield of the photosynthetic activtity (Y(II)) and reconstruction of light absorbing pigments, were analyzed over three days (Figure 3).

During the course of resuscitation, the photosynthetic activity of the WT recovered at the same pace as the Δ*phaEC* mutant (Figure 3A). Likewise, there was no difference in the re-appearance of photosynthetic pigments, indicated by the absorption at 630 nm and 680 nm (Figure 3B and Figure 3C, respectively). The full spectra between 600 and 750 nm are shown in Figure A1.

To test if the formation of PHB is beneficial for chlorotic cells under conditions of certain abiotic stresses, WT and Δ*phaEC* cells were nitrogen starved for two weeks to induce PHB production. Subsequently, one of the following experimental setups was performed: (1) One specific abiotic stress was applied to the chlorotic culture for a specific time, before the culture was plated on BG_11_ agar plates and the formation of CFU was determined. (2) Chlorotic cells were plated on BG_11_ plates, which contained additional components causing stress (for example, higher salt concentrations). The results are summarized in Table 1.

Under all tested conditions, no growth advantage was observed for the WT compared to the Δp*haEC* mutant strain. Some conditions were too harsh for any cells to survive—for example, the simulation of heat at 50 °C. To illustrate what a typical result looked like, a representative drop plate assay is depicted in Figure 4. In this example, no viability difference between WT and Δ*phaEC* cells was observed after resuscitation from prolonged chlorosis and growth on 100 mM NaCl. In summary, there was no condition found where the possession of PHB was advantageous for the WT.

### 3.3. Heterogeneity of PHB Production

In microscopic studies, we noticed that the number of PHB granules varied strongly between individual WT cells during nitrogen starvation. While most cells did contain PHB, the amount varied greatly, both in the size as well as in the number of PHB granules (Figure 5).

Since this observation has not been systematically addressed before, we further investigated this phenomenon via flow cytometry (FC). Therefore, *Synechocystis* cells were starved for two weeks from nitrogen and stained with Bodipy. To distinguish the fluorescence signal of Bodipy stained PHB from background signals, two controls were performed: (1) determination of the unspecific background fluorescence from unstained WT cells as well as (2) the fluorescence of Bodipy stained PHB free Δ*phaEC* cells (Figure 6).

Compared to the unstained WT cells, the Δ*phaEC* cells showed a higher fluorescence signal, which is caused by unspecific staining of hydrophobic structures within the cells by Bodipy. Compared to the Bodipy stained Δ*phaEC* cells, a large portion of the Bodipy stained WT cell showed a fluorescence signal that was partially overlapping with that of Δ*phaEC* cells, but, on average, shifted to higher intensities (Figure 6, blue circle). This corresponds to a major population of cells that contained only low to medium amounts of PHB. From these, a second part of the WT population could clearly be distinguished, which showed much higher fluorescence signals (Figure 6, brown circle). This corresponds to a subpopulation of high PHB producing cells. Since the cells used for PHB production are derived from a single clone, they are assumed to be genetically identical. To definitively clarify that the different PHB synthesis phenotypes are not caused by (epi)genetic differences but are more likely, and are based on stochastic regulation of PHB synthesis, we isolated low and high PHB producing cells by FACS sorting. Single cells were recovered and grown on BG_11_ agar plates until colonies appeared. Several colonies derived from high- or low-producing cells were separately pooled and used to inoculate a fresh BG_11_ culture. Afterwards, the cultures were shifted again to nitrogen free BG_0_ medium to trigger PHB synthesis and the cells were again investigated using FC as described above. The results are shown in Figure 7.

Undoubtedly, the cells in these new cultures established again the same PHB heterogeneity as in the previous experiment, regardless if they were derived from previously low- or high-producing cells. As previously described, all Bodipy stained WT cells showed a higher overall fluorescence compared to the unstrained control and the Bodipy stained ΔphaEC mutant strain, indicating that most cells did contain PHB. Furthermore, all replicates showed the characteristic shoulder in the FC analysis, representing cells with high PHB contents. This result demonstrates that the regulation of PHB synthesis in nitrogen-starved cells follows a stochastic program, resulting in a mixed population with a majority of low- to medium PHB producers and a minority of high PHB producing cells.

## 4. Discussion

### 4.1. Heterogeneity of PHB Content

When living in a changing environment, being prepared for different potential outcomes can be beneficial. In order to be prepared for unpredictable future scenarios, many bacteria have evolved a strategy of phenotypical heterogeneity for bed hedging [24]. This allows them to have a certain part of their population being well prepared for either outcome. In this work, we could show that the formation of PHB in *Synechocystis* also represents this form of phenotypical heterogeneity. We observed this via fluorescence and electron microscopy (Figure 5) and further confirmed it by flow-cytometry (Figure 6). As already hypothesized, this heterogeneity has no inheritable reason, since cells that were separated into low- and high producers produced progeny with the same heterogeneity in PHB content as the ancestral generation (Figure 7). It is likely that the heterogeneity results from a probabilistic genetic program (bet-hedging) that results in the observed distribution of PHB production. The expression of key enzymes (such as PhaEC) might be quite variable, resulting in some cells that do produce more PHB than others. It was shown before that transcriptional infidelity promotes heritable phenotypic changes, which could also explain the PHB heterogeneity [25]. The phenomenon of a bet-hedging strategy in PHB contents has also been described in a different manner for some heterotrophic bacteria, such as *Sinorhizobium meliloti.* Here, when dividing *S. meliloti* cells face starvation, they form two daughter cells with different phenotypes, one with low and one with high PHB content. These daughter cells are adapted to either short- or long-term starvation, respectively [26].

### 4.2. Physiological Function of PHB

Since the majority of cells contained only minor or medium amounts of PHB, it has to be assumed that the conditions, under which it is beneficial to contain large PHB quantities, are rather seldom. To better understand PHB metabolism, this work aimed at providing additional information about when and why PHB is formed. We show clear evidence that the PHB content within the cells is increased when the bacteria are grown under alternating day–night rhythm (Figure 1, Figure 2). This is in coherence with a recently published transcriptomic data set [27]. Here, the authors have analyzed the transcription of genes under two consecutive dark- and light phases. We extracted all known PHB related genes from this data set and found a clear correlation between the genes of PHB metabolism and the diurnal rhythm (Figure A3). The assumption that PHB might play a role during dark, anaerobic conditions was already hypothesized before [28].

Since PHB is formed from intracellular glycogen pools [17] and cyanobacteria catabolize glycogen during the night [29], this phenotype might be explained by an increased carbon availability from glycogen during the night. Furthermore, *Synechocystis* accumulated less PHB when grown under standing conditions (Figure 1). These findings are in line with previous reports, which showed a generally decreased metabolic activity under conditions of limited gas-exchange and CO_2_ availability [30]. In addition, photosynthetically active *Synechocystis* cells produce oxygen, which accumulates in the medium when grown under standing conditions (Figure A2). Excess of oxygen is known to impair the efficiency of the RuBisCO enzyme by causing the oxygenase reaction, which likely resulted in a reduced PHB reduction due to the carbon loss caused by photorespiration.

The results fit to the observation that the EMP pathway proved to be the most important carbon pathway for PHB production (Figure 2). Studies have shown that cyanobacteria employ the EMP pathway for degradation of glucose residues to pyruvate [30]. This is also the case for growth in dark phases [31]. Furthermore, the EMP produces less NADPH and more ATP, compared to other glycolytic routes, such as the OPP (oxidative-pentose-phosphate pathway) or the ED (Entner-Doudoroff) pathway. Although the cultivation under standing conditions is not a strictly anaerobic condition (Figure A2), the O_2_ limitation during the dark phase could be sufficient to induce fermentation-like processes. It is known that cyanobacteria carry out fermentation under dark/anoxic conditions and produce a variety of different fermentation products [32]. Although PHB is so far not considered a cyanobacterial fermentation product, these conditions of limited oxygen availability and absence of light could explain the observed increase of intracellular PHB (Figure 1).

In contrast to the common belief, the ability to produce PHB under conditions of nitrogen starvation was not shown to be beneficial since we could not detect any physiological differences between WT and Δ*phaEC* cells during the resuscitation process (Figure 3). Even when additional abiotic stresses were added, no growth advantage was observed (Table 1). It might be that growth under the controlled environment of laboratory conditions is not limiting the cells, neither in carbon nor energy, and hence no phenotypes were visible.

It was previously shown that excess NADPH under nitrogen-starved conditions sustains PHB accumulation [13]. Hence, PHB might serve as an intracellular pool for electrons under conditions of excess reduction equivalents. The fact that we did not observe any phenotypical difference under conditions of electron excess (for example under high amounts of light). 

Table 1 might be explained by other regulatory mechanisms compensating for the lack of PHB. Since the correct regulation of intracellular redox-state is crucial for the cell physiology, *Synechocystis* has evolved various strategies to cope with high levels of reduction equivalents. One example for such a mechanism is the flavodiiron protein Flv3, which serves as a sink for excess electrons from the photosynthetic light reaction, by converting O_2_ to H_2_O. A recent publication showed indeed increased PHB synthesis in a Flv3 deficient strain [33]. If PHB is indeed serving as an intracellular electron sink, its absence might be compensated by a higher activity of Flv3. Alternative ways of getting rid of reduction equivalents could be the secretion of reduced organic molecules, such as acetate. It can be assumed that the correct regulation of the ATP/NADPH ratio is crucial for cyanobacterial cells. Since the production of PHB from glycogen provides ATP but consumes NADPH, the biopolymer could help the cells to regulate this ratio under conditions of electron excess [33].

## 5. Conclusions

This work describes different factors that influence the formation of PHB. However, the conditions where PHB is advantageous during nitrogen starvation have yet to be discovered. The results of this work can help to create strains with enhanced PHB contents. Besides applying dark phases, finding regulators of the EMP pathway to unlock the carbon flow from glycogen to PHB could further boost the production [17]. One alternative strategy could be the deletion of other NAPDH consuming pathways, as already shown in a Δflv3 strain. Finally, a deeper understanding of the PHB-heterogeneity might result in a more homogenous culture of high-producing cells, which is beneficial for the overall yield.

## Figures and Tables

**Figure 1 life-10-00047-f001:**
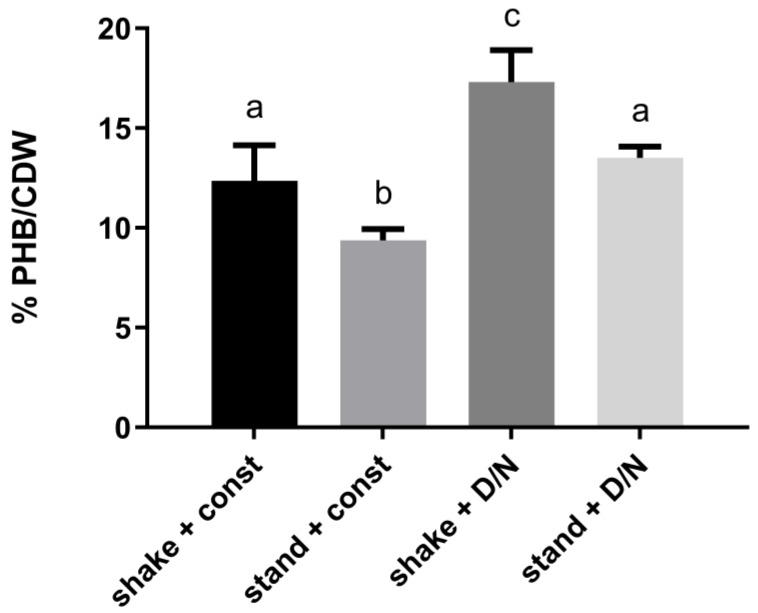
Quantification of polyhydroxybutyrate (PHB) content after 13 days of nitrogen starvation. WT cells were grown under different conditions of aeration and illumination. All cells were pre-adapted to these conditions three days before the shift to nitrogen starvation. Shake = continuous shaking at 120 rpm. Stand = cultures were standing without any shaking. Const = constant illumination with ~50 µE. D/N = altering illumination with 12 h of light (50 µE) and 12 h dark. Data shown as mean ± SD of three biological experiments; levels not connected by the same letter are significantly different (*p* ≤ 0.05). CDW = cell dry weight.

**Figure 2 life-10-00047-f002:**
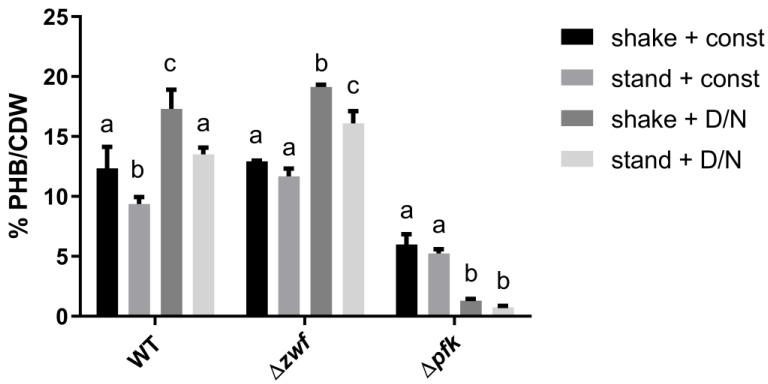
Quantification of PHB content after 13 days of nitrogen starvation. Mutant strains lacking either the EMP (Δ*pfk*) or the OPP and ED pathway (Δ*zwf*) were compared to the WT strain to test the influence of these carbon pathways on the PHB production. The cells were grown under different conditions of aeration and illumination. Shake = continuous shaking at 120 rpm. Stand = cultures were standing without any shaking. Const = constant illumination with ~50 µE. D/N = altering illumination with 12 h of light (50 µE) and 12 h dark. Data shown as mean ± SD of three biological experiments; levels not connected by the same letter are significantly different (*p* ≤ 0.05; only within the same genetic background).

**Figure 3 life-10-00047-f003:**
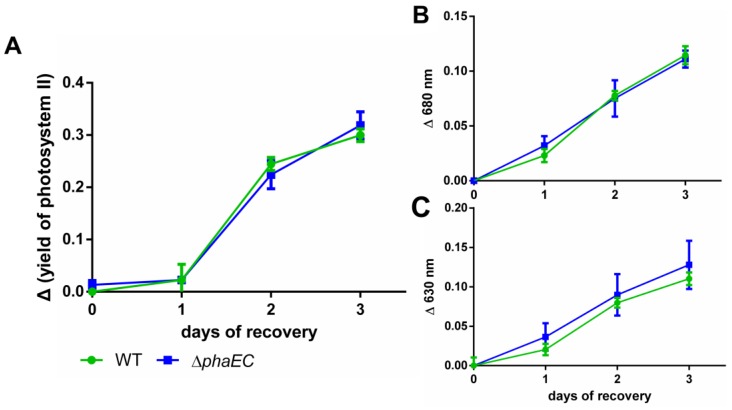
Physiological parameters during the resuscitation after 18 days of nitrogen starvation. Cells were grown under alternating day-night rhythm and continuous shaking; (**A**) PAM measurements to determine the maximum PSII quantum yield (Fν/Fm). The measurements were normalized to the yield at timepoint 0; (**B**,**C**): difference in absorption of normalized spectra during resuscitation. Spectra, which were normalized to OD_750_ (Figure A1), were used to calculate the difference between the initial absorption values (at wavelength of 630 and 680 nm) at time point 0 (day 0) and the various time points during resuscitation. All samples represent three individual biological replicates.

**Figure 4 life-10-00047-f004:**
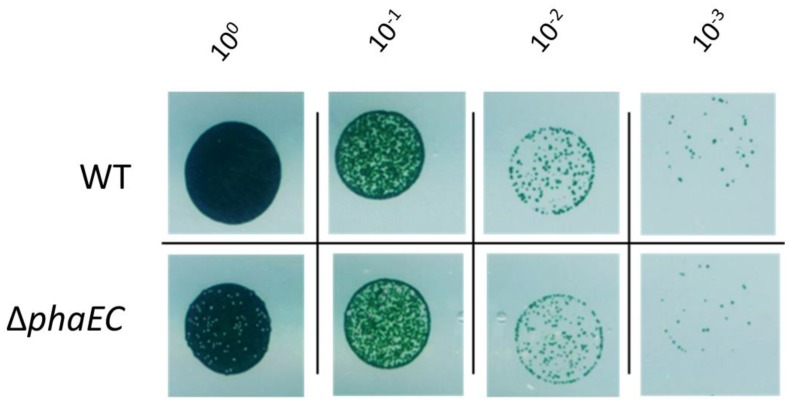
Viability assay of *Synechocystis* WT and Δp*haEC* mutant cells using the drop plate method after two weeks of nitrogen starvation and subsequent growth in BG_11_ plates with 100 mM NaCl. Several dilution steps were dropped on BG_11_ agar plates, ranging from OD_750_ = 1 (represents dilution 10^0^) to OD_750_ = 0,001 (represents dilution 10^−3^).

**Figure 5 life-10-00047-f005:**
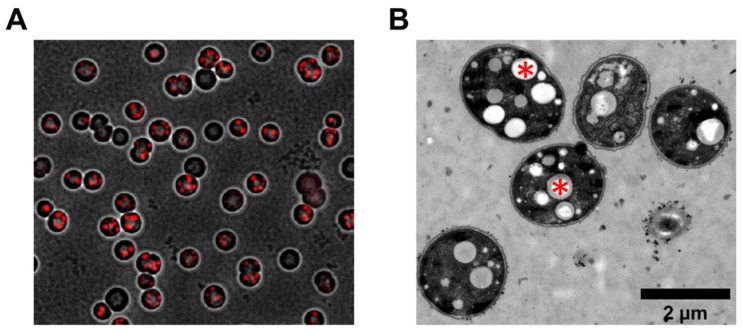
Microscopic analysis of varying PHB contents in WT cells; (**A**) fluorescence microscopy of WT cells after 14 days of nitrogen starvation. PHB granules are visualized by staining with Nile red; (**B**) TEM picture of WT cells after 17 days of nitrogen starvation. Representative PHB granules are indicated in two different cells by red asterisks.

**Figure 6 life-10-00047-f006:**
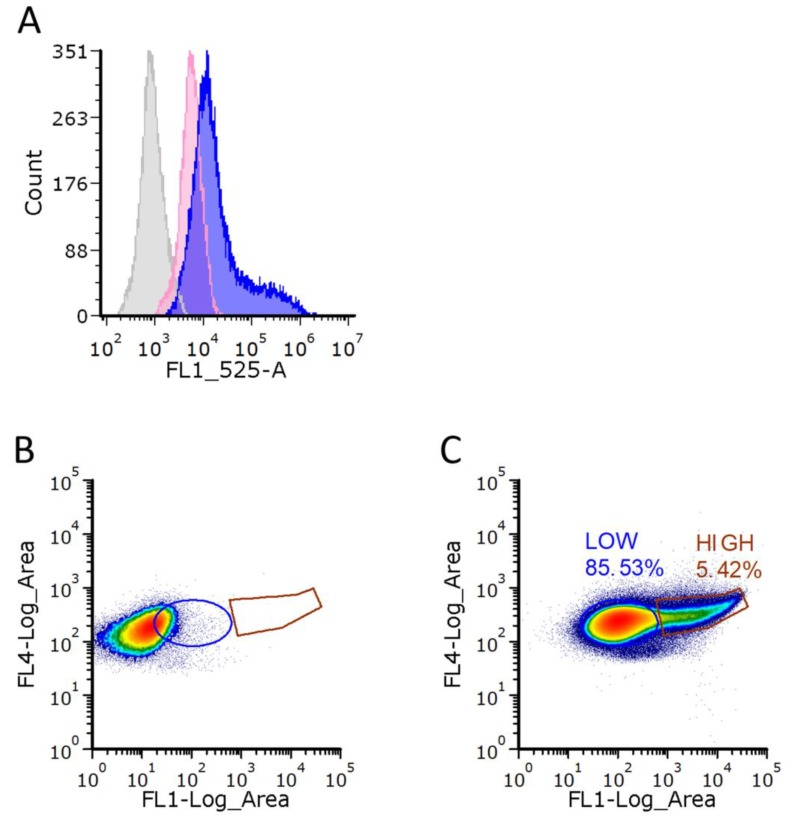
Analysis of intracellular PHB content using FC to detect *Synechocystis* cells stained with Bodipy. (**A**) overlay of the measurements for unstained WT (grey), Δp*haEC* mutant strain stained with Bodipy (red) and WT stained with Bodipy (blue). To illustrate how the cells were separated with FACS, the sort regions for Δp*haEC* mutant strain and WT are shown in (**B**,**C**), respectively. The red peak of Δp*haEC* cells in (A) corresponds to the cell population in (B), while the blue peak of WT cells in (A) corresponds to the cell population in (C). Sort regions in WT for low- and high producers are indicated as a blue or brown circle, respectively, in (B) and (C). FL4 = Chlorophyll emission and FL1 = Bodipy emission.

**Figure 7 life-10-00047-f007:**
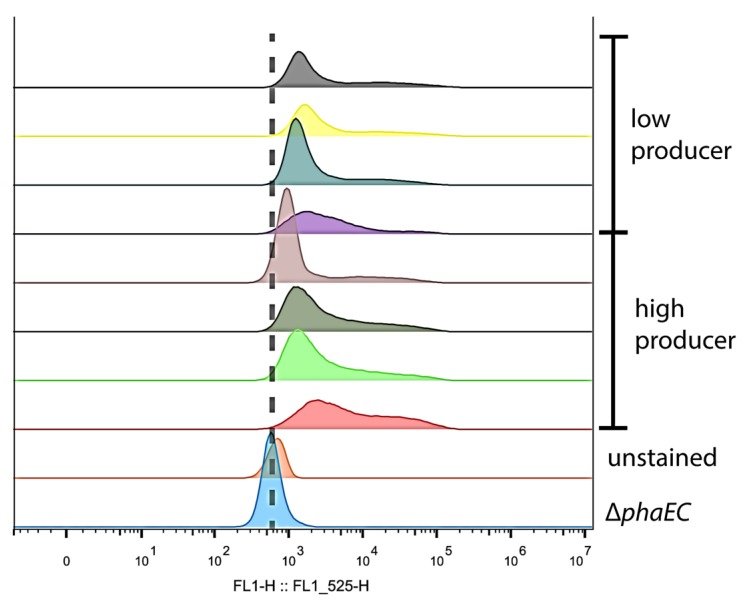
FC analysis of intracellular PHB content. “Low” and “high” represent four individual biological replicates from FACS, which were based on cells that contained low or high amounts of PHB, respectively. Cells were starved from nitrogen for two weeks and stained with Bodipy (except for the unstained control).

**Table 1 life-10-00047-t001:** List of abiotic stresses applied to WT and Δ*phaEC* cells, which were starved for two weeks of combined nitrogen sources. A detailed description is listed under “Treatment”. For the first category of experiments, cells either treated for a specific time with the conditions listed under (1). After the treatment, cells were grown on BG_11_ agar plates according to the drop plate method as depicted in Figure 4. Alternatively, chlorotic cells were transferred to BG_11_ plates containing the ingredients listed under category (2) to apply the abiotic stress during the recovery process. Observed differences are listed under “Effect”. When “no difference” was observed, both strains (WT and Δ*phaEC*) showed similar amounts of colonies. “All cells were dead” indicates experiments, where no colonies appeared. All treatments were tested in three different biological replicates.

Abiotic Stress	Treatment	Effect
**(1)**		
**Cold**	4 °C over night 1 h at −20°C; 14 h at −20 °C	No difference
**Heat**	45 min at 40 °C; 45 min at 40°C and 20 min at 50°C	No difference All cells were dead
**Physical Force**	Centrifugation for 30 min at 20,000 g; 1 h at 25,000 g 3 × 5 min at 4 m/s glass-bead milling in Ribolyser; 3 × 5 min at 7 m/s Ribolyser	No difference No difference All cells were dead
**Darkness**	1 day; 2 days, 5 days; 8 days; 10 days; 15 days	No difference
**Low Light**	14 days at 5 µE	No difference
**Alternating Light**	12/12 light/dark	No difference
**Drought**	30 min at 30 °C SpeedVac	All cells were dead
**Nitrogen Starvation**	3 weeks 10 weeks	No difference ∆*p**haEC* showed weak growth advantage
**High Light**	1 day incubation at 500 µE	No difference
**(2)**		
**Buffered Medium**	BG_11_ agar plates containing 300 µL, 1 mL, 3 mL TES buffer	No difference
**Salt**	Recovery at BG_11_ agar plates with: 100, 150, 300 mM NaCl 2 × BG_11_ salts	No difference No difference
**Carbon Availability**	0, 10, 50, 150 mM bicarbonate (added to BG_11_ agar plates)	No difference

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
