# Peer review of "On the Role and Production of Polyhydroxybutyrate (PHB) in the Cyanobacterium *Synechocystis* sp. PCC 6803"

_life, 2020, doi:10.3390/life10040047_

Round 1

Reviewer 1 Report

This paper describes the physiological conditions that are important for polyhydroxybutyrate (PHB) synthesis a possible polimers used as bioplastic. Authors use a model cyanobacteria Synechocystis that bearing genes required for PHB synthesis but the physiological function of this compound remains unknown.
Authors demonstrate that the accumulation of intracellular PHB was higher when cyanobacterial cells were grown in alternate day-night in nitrogen starvation conditions. But it's not clear if this accumulation is similar in limited nitrogen conditions. This is a key point because changes in  C/N ratio determined the production of (PHB). The authors used BG11 medium with 5 mM of NaHCO3 in nitrogen starvation conditions (line 75). What happened if you used different % of CO2 to bubble a BG11 nitrogen starvation culture??
Line 152 illumination
Line 159. Related with Figure 1. Are there differences with PHB production at  0, 24 or 48h before nitrogen starvation?
Line 166. Figure A2 is referred to as oxygen measurement in standing conditions?? What happened with oxygen in shake conditions?
In the case of standing cultures, I think that in these conditions also affect to cellular viability. Then It is necessary to measure the number of viable cells (by counting with FC )
 Line 195. It is very interesting results related to pfk mutant strain.
Line 211. During resuscitation, I have missed the PHB measurement. It’s a key point to evaluate if this compound is important for this process.
Line 217. What % of PHB/CDw is measured in these conditions? And after darkness???
Line 232. The result obtained from FC revelaled a wide distribution of the PHB content and different cell size. This data is important for the quantification of PHB content that is produced in the different conditions. If we take this into account, perhaps the differences found Light/dark cycles would be different.
Line 314. The authors did not detect physiological differences during the resuscitation process in continuous light but in L/D conditions???
I don’t understand why the author doesn’t use figure A1 in the paper. It is important to note that there are differences in pigments in PhaEC mutant high ficolbiliproteins and carotene…Is it probably it is related to low PHB production???
I think that is necessary quantification of PHB content through the resuscitation process to conclude that this compound is not necessary.

Author Response

Dear reviewer 1,

please find attached a summary of all your comments and our responses. Thanks in advance for thoroughly checking our manuscript and your constructive feedback.

Kind regards,
Moritz Koch (for all the authors)

Reviewer 2 Report

Dear Authors,

The MS called « On the role and production of PHB in the cyaobacterium Synechocystis PCC6803 » tends to evaluate different abiotic conditions in increasing the polyhydroxybutyrate production in one cyanobacteria WT and its mutants.  However a lot of issues can be made about major lacks of details in material and methods, an erratic organization throughout the text, with some « mix » of methodological details in results, and a discussion which repeated the results section… the organization with clearly separate sections and with removal of repetitions (between results and discussion) are required to improve the MS. Besides, the absence of statistical analyses can be deplored in the study as well, as some comparison between WT and mutants facing to several treatments might confirm the observations and highlight the purpose. The obtention of the different mutants are lacking in Mat & Meth and must be precisely described in this part. Only the mutant ΔPhaEC is mentioned in Mat & Meth while in fig.2 there are Δzwf & Δpfk… several improvements and changes can be made in all tables and figs as well, and the text and several sentences have to be more focused on results and more precise. The english text has to be carefully checked by a native speaker.

Minor corrections

  • No abbreviation in the title, « PHB » has to be entirely written and understanding by the large audience
  • Abstract : remove the Synechocystis in ( ) as you talked about this strain already. Add the vocable (cyanobacterium) instead. The abstract is not so informative and need to be focused on the objective of the study and why the PHB is important to investigate here.
  • Introduction : is the « redoxstress » correct (lin53) ?
  • The text must be aligned in both sides throughout the text, please check out.
  • Mat & Meth : Please add a section to consider the obtention and use of all mutants investigated in thi study. How do you get them ?
  • More details about the « physical stress conditions », including the different time of incubation, each experiment, replicates…
  • Sections « microscopy and staining procedures » and « Electron microscopy » must be grouped together… « oxygen measurements » can be placed with physical stress conditions… the methods and tools must be better organized according to the domains of specificity and not in hazardous positions…
  • Some statistical analyses just simple tests (ANOVA…) could be more informative than only observations when comparisons are made between treatment vs standard an so on..
  • Results :
  • The title 3.1 is rather unclear, and need to be more precise…
  • The part from lines 176-182 is rather informative about the methodology, and could be placed in Mat & Meth and not here.
  • Fig2 in legends you have to specify the differences in mutants used in this graph.
  • Why don’t you use the growth curves (OD750nm) during nitrogen starvation to assess the physiology of each strain faced to the stress (comparison between mutants & WT, vs optimal conditions vs N stress ?
  • Fig3 : Precise the axis title «ΔY(II))  please.
  • The B & C graphs are too small or be placed as suppl data.
  • The transition sentences (lines 215-219) are not needed here. Please remove all these transitions at the beginning of each section. They are not really need.
  • Table 1 : is unprecise and need more information and statistical analyses. Effect with « no difference » or « all cell were deads » without confirmation, no value nor graph are insufficient here. Besides where are all the treatments realized in this table ? no other figure nor explanation…
  • Line 233 : « mircoscopic »
  • The fig 4a, is not visible and must be more contrasted or with bigger size. The Fig4 B : some phenotypic distinctions (like granules or other features) must be shown directly on the picture, by symbols or arrows..
  • Fig5 : too small and specify the axis title FL4 log area ?
  • The legends of Fig 5 are not so clear especially to explain B & C
  • The section in lines 264-271 is not clear, what do you mean exactly ? why re-inoculate the cells ?
  • Fig 6 is not visible, the peak has to be more contrasted or larger. Please improve it. What the meaning of this graph indeed ?
  • Discussion : too close to the results part, including artificial transition sentences and repetitive from the previous part (lines 321-324, lines : 341...). Why not placing the heterogeneity of PHB content before the physiological function and experimental stress response ?
  • A conclusion could be useful to synthetize the results obtained and open to some industrial or perspective to highlight the interest of increasing the PHB production in cyanobacteria.
  • References :

Check carefully all the references, if this review used the abbreviated or the complete names of each review, as both are found here.

Provide itaclics for each species name please.

Author Response

Dear reviewer 2,

please find attached a summary of all your comments and our responses. Thanks in advance for thoroughly checking our manuscript and your constructive feedback.

Kind regards,
Moritz Koch (for all the authors)

Round 2

Reviewer 1 Report

Authors has answered my questions satisfactorily and made the appropriate changes to the text.

There are one more suggestions:
line 123 PAM measureaments. More detail is needed in this section. It is not described if the cells have been in darkness and how long and how Y(II) is determined.
Figure 3. PSII operating efficiency [Y(II)] (calculated as (Fm′–Fs)/Fm′) or PSII maximum efficiency (Fv′/Fm′)calculated as (Fm′–F0′)/Fm′????

Author Response

Dear Reviewer I, 

please find our comments attached.

Best regards, 

Moritz Koch (for all authors)

Reviewer 2 Report

Dear Authors,

Thank you for sumitting this new brand version, much more visible, with figures more easiily to read, bigger size and with colors. You included as well some statistical data wich were lacking in the primer version. You delete some redundances betwenn the results and discussion and rewritte some parts of the MS.

. The english was corrected, heavy and unclear sentences rewritten, and all the references were ckecked as well.

I can only accept this efforts and this new MS more attractive and easy to understand for a large audience

Author Response

Dear Reviewer II, 

thank you very much for thoroughly reading through our revised manuscript and your constructive feedbrack throughout this review process. We highly appreciate it since it drastically improved the appearance and content of the final mansucript. 

Best regards, 

Moritz Koch (for all authors)